

# Assessment of medical information on irritable bowel syndrome information in Wikipedia and Baidu Encyclopedia: comparative study

Xi Li[1], Kexin Chen[2,3,4], Yongbin Jia[2], Fang Yin[2,3,4], Xi Wen[5], Chunhui Wang[3], Zhipeng Li[1] and Hu Zhang[2,3,4]

[1] General Practice ward/ International Medical Center Ward, General Practice Medical Center, West China Hospital of Sichuan University, Chengdu, Sichuan, China
[2] Lab of Inflammatory Bowel Disease, Frontiers Science Center for Disease-Related Molecular Network, Chengdu, Sichuan, China
[3] Department of Gastroenterology, West China Hospital of Sichuan University, Chengdu, Sichuan, China
[4] Centre for Inflammatory Bowel Disease, West China Hospital of Sichuan University, Chengdu, Sichuan, China
[5] Department of Urology, West China Hospital of Sichuan University, Chengdu, Sichuan, China

Corresponding authors
Zhipeng Li, 13568985243@163.com
Hu Zhang, zhanghu@scu.edu.cn

## ABSTRACT

**Background**. Irritable bowel syndrome (IBS) is a functional gastrointestinal disorder (FGID) with heterogeneous clinical presentations. There are no clear testing parameters for its diagnosis, and the complex pathophysiology of IBS and the limited time that doctors have to spend with patients makes it difficult to adequately educate patients in the outpatient setting. An increased awareness of IBS means that patients are more likely to self-diagnose and self-manage IBS based on their own symptoms. These factors may make patients more likely to turn to Internet resources. Wikipedia is the most popular online encyclopedia among English-speaking users, with numerous validations. However, in Mandarin-speaking regions, the Baidu Encyclopedia is most commonly used. There have been no studies on the reliability, readability, and objectivity of IBS information on the two sites. This is an urgent issue as these platforms are accessed by approximately 1.45 billion people.

**Objective**. We compared the IBS content on Wikipedia (in English) and Baidu Baike (in Chinese), two online encyclopedias, in terms of reliability, readability, and objectivity.

**Methods**. The Baidu Encyclopedia (in Chinese) and Wikipedia (in English) were evaluated based on the Rome IV IBS definitions and diagnoses. All possible synonyms and derivatives for IBS and IBS-related FGIDs were screened and identified. Two gastroenterology experts evaluated the scores of articles for both sites using the DISCERN instrument, the Journal of the American Medical Association scoring system (JAMA), and the Global Quality Score (GQS).

**Results**. Wikipedia scored higher overall with DISCERN ($p < .0001$), JAMA ($p < .0001$) and GQS ($p < .05$) than the Baidu Encyclopedia. Specifically, Wikipedia scored higher in DISCERN Section 1 ($p < .0001$), DISCERN Section 2 ($p < .01$), DISCERN Section 3 ($p < .001$), and the General DISCERN score ($p < .0001$) than the Baidu Encyclopedia. Both sites had low DISCERN Section 2 scores ($p = .18$). Wikipedia also had a larger percentage of high quality scores in total DISCERN, DISCERN Section 1, and DISCERN
Section 3 ($p < .0001$, $P < .0001$, $P < .0004$, respectively, based on the above 3 (60%) rule).

**Conclusions.** Wikipedia provides more reliable, higher quality, and more objective IBS-related health information than the Baidu Encyclopedia. However, there should be improvements in the information quality for both sites. Medical professionals and institutions should collaborate with these online platforms to offer better health information for IBS.

## INTRODUCTION

The Internet has revolutionized our access to health information and medical knowledge (*Ma et al., 2020*). As of June 30, 2022, there were 7.93 billion Internet users worldwide, with 18.26% of them (1.45 billion) located in China (*Internet World Stats, 2023a*; *Internet World Stats, 2023b*). About 80% of online platform users in the United States regularly used the Internet to search for medical information (*Basch et al., 2017*). Almost 28.5% of Internet users in China (300 million) have used paid online medical services (*Yunbox, 2022*). Internet searches are now the preferred way for people to obtain information on health-related issues (*Eysenbach, 2009*). However, many online sources are inaccurate and misleading (*Swire-Thompson & Lazer, 2020*; *Suarez-Lledo & Alvarez-Galvez, 2021*). Online health information should be evaluated for credibility and veracity as soon as possible.

Wikipedia is the most widely used online encyclopedia, with over six million articles in English, including 30,000 articles on medical topics (*Shafee et al., 2017*; *Wikipedia*, 2023a). These medical articles get more than 10 million views per day (*Heilman & West, 2015*; *Laurent & Vickers, 2009*). The Baidu Encyclopedia is the leading online encyclopedia in China, with more than 26 million articles in Chinese (*Baidu, 2023*). Health and medical topics were the most searched subjects among 1,500 science topics, with a proportion of 53.1% of all searches (*Yunbox, 2022*).

Irritable bowel syndrome (IBS) is a type of functional gastrointestinal disorder (FGIDs) that causes abdominal pain, altered stool consistency or frequency, and is diagnosed by a number of atypical symptoms. These symptoms can impair the quality of life and social functioning of the affected individual (*Lacy et al., 2016*; *Ford et al., 2020*). The global prevalence of IBS varies from 1.3% to 7.6% (*Lovell & Ford, 2012*), while in mainland China it ranges from 1.4% to 11.5% (*Study Group of Functional Gastrointestinal Disorders, Study Group of Gastrointestinal Motility, Chinese Society of Gastroenterology, Chinese Medical Association, 2020*). FGIDs are a group of idiopathic disorders that affect different parts of the digestive system, which include IBS, functional dyspepsia, and functional constipation (*Black et al., 2020*).

Intriguingly, IBS symptoms vary widely, with gastrointestinal symptoms including abdominal pain, diarrhea, constipation, bloating, and feelings of fullness (*Ford et al.,*

*2008*). There is a confusion between the symptoms in IBS and some FGIDs as they develop over time (*Halder et al., 2007*). According to the Rome IV criteria, 18 of 33 FGIDs have symptoms that overlap with IBS and are grouped into four themes by anatomical location: (1) IBS, which is categorized by the main stool pattern; IBS-C, IBS-D, IBS-M, and IBS-U; (2) gastroduodenal disorders (GDs), such as functional dyspepsia and belching disorders; (3) bowel disorders (BDs), such as functional diarrhea, functional constipation, and functional abdominal bloating; and (4) anorectal disorders (ADs), such as fecal incontinence, functional anorectal pain, and functional defecation disorders. For diseases with gastrointestinal symptoms like abdominal pain, diarrhea, constipation, and bloating, the concept of FGIDs is more scientific, precise and comprehensive from a professional perspective.

However, IBS is more often searched online than other FGIDs, due to its diagnostic features, high incidence, and ease of understanding the material. The quality of online information on IBS is important for people who seek self-diagnosis and self-treatment based on the internet (*Cruz et al., 2019*). However, few studies have evaluated the reliability, readability, and objectivity of online IBS information (*Levine et al., 2020*). This study aimed to assess the IBS information found in the Baidu Encyclopedia and Wikipedia regarding IBS and some FGIDs (GD, BD, and AD), which present the same symptoms as IBS, to help people find trustworthy information, avoid misinformation and misdiagnosis, and promote the development of Internet medicine.

## METHODS

### Data sources

The articles analyzed in this research were acquired from the English Wikipedia and Chinese Baidu Encyclopedia on December 21, 2022. Wikipedia and the Baidu Encyclopedia were searched for articles based on Rome IV Criteria (*Drossman & Hasler, 2016*),  which are globally recognized guidelines for diagnosing and classifying IBS and other FGIDs (*Rome Foundation, 2023*). All possible synonyms and derived words for every term were searched. Two reviewers with decades of experience in gastroenterology and competence in the diagnosis and treatment of IBS and FGIDs evaluated these articles from Baidu Encyclopedia and Wikipedia. If there were any disagreements, the third examiner reviewed and arbitrated the dispute.

### Assessment of the quality of the research articles

The quality of disease-related articles was assessed using the JAMA scoring system (*Silberg, Lundberg & Musacchio, 1997*), the DISCERN instrument (*Charnock et al., 1999*), and GQS (*Bernard et al., 2007*). Appendix S1 displays the specific contents of these scoring instruments. The DISCERN instrument evaluates health information quality. It has 15 questions, which are divided into three sections: reliability (Section 1), treatment choices (Section 2), and overall quality (Section 3). Each question is scored from 1 to 5. The total scores can range from 16 to 80. A score of 63–80 means excellent quality, 51–62 indicates good quality, 39–50 is fair quality, and 16–38 indicates poor quality results (*Charnock et al., 1999*). We set the cutoff value at 3 for each question, based on previous research, a score

above 3 (60%) indicated good quality (*Ma et al., 2022*). The JAMA scoring system assesses the quality of health-related websites based on four criteria: author, attribution, disclosure, and currency. The GQS assesses each website's overall quality with a subjective measure. It uses a 5-point scale to evaluate how easy and smooth the website is to use (*Bernard et al., 2007*).

## Statistical analysis

We checked data normality using the Shapiro–Wilk test. We reported descriptive analyses as means and SDs for normally distributed variables, and as medians and IQRs for non-normally distributed ones. We assessed interobserver reliability with the intraclass correlation coefficient (ICC), which ranges from 0 (untrusted) to 1 (fully trusted), and discussed any values below 0.75 with the research team. We compared different ranks with the Mann–Whitney U test for nonparametric tests using SPSS 22.0. We used Fisher's exact test to compare the frequency distribution of the DISCERN scores. We performed statistical analyses and created figures with Prism 8.4.0 (GraphPad Software, San Diego, CA, USA). $P < 0.05$ was as the significance level for the statistical analysis.

# RESULTS

## Overall scores for the Baidu Encyclopedia and Wikipedia

To assess the inter-rater reliability of the two independent reviewers, a 2-way mixed/random effects model was applied. Ratings for GQS, JAMA, and DISCERN scores were highly consistent between the reviewers, with ICC values of 0.759, 0.955, and 0.759, respectively.

The scores for total diseases (containing IBS, GD, BD, and AD) showed that Wikipedia outperformed the Baidu Encyclopedia in terms of quality, readability and usability of health information; we applied three different tools: the DISCERN, JAMA, and GQS to assess the contents of both sites (Fig. 1A). The DISCERN tool consists of three sections: reliability, treatment choices, and overall rating. Wikipedia scored significantly higher than the Baidu Encyclopedia in all three sections, indicating that it provides more reliable and objective information, better guidance for patients' decision making, and higher overall quality of the publication (Fig. 1B). The JAMA and GQS tools also indicated that Wikipedia had better reading fluency and was easier use than the Baidu Encyclopedia. The DISCERN General score reflected the overall quality of the websites. Wikipedia scored significantly higher, suggesting that it is a more trustworthy and helpful source of health information.

## Overall quality comparison for the theme of IBS between Wikipedia and the Baidu Encyclopedia

The classical symptoms of IBS include abdominal pain and bloating associated with a change in stool form or frequency. It is divided into four subtypes by the fecal trait: IBS-C, IBS-D, IBS-M, and IBS-U. Wikipedia scored higher in the total DISCERN score on IBS articles, but there were no obvious discrepancies in the JAMA and GQS scores (Fig. 2A). Wikipedia scored higher according to DISCERN Sections 1, 2, 3, and the General section (Fig. 2B). These results indicate a notable difference between the two sites on IBS content.

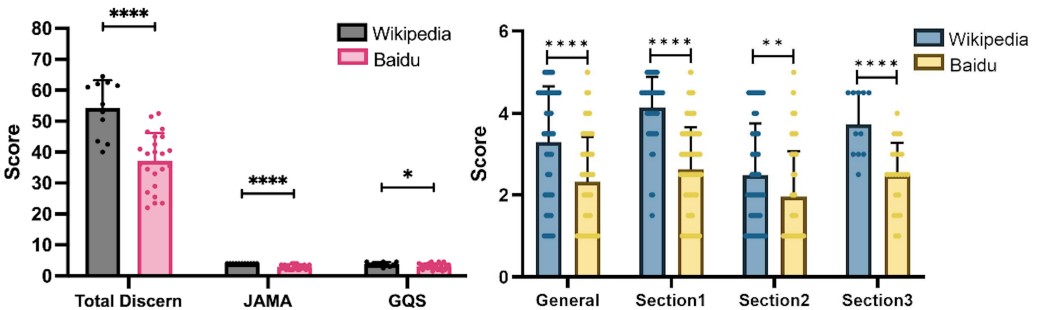

**Figure 1  Scores for Baidu Encyclopedia and Wikipedia articles for all diseases.** (A) Total DISCERN, JAMA, and GQS scores from the Baidu Encyclopedia and Wikipedia articles. (B) The general and 3 DISCERN section scores. * $p < 0.05$, ** $p < 0.001$, *** $p < 0.0001$, ns $p > 0.05$.

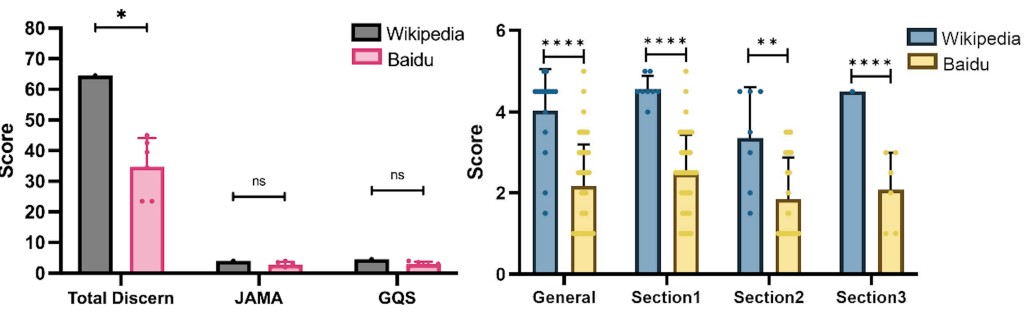

**Figure 2  Scores from the Baidu Encyclopedia and Wikipedia articles for IBS.** (A) Total DISCERN, JAMA, and GQS scores from the Baidu Encyclopedia and Wikipedia articles. (B) The general and 3 DISCERN section scores. * $p < 0.05$, ** $p < 0.001$, *** $p < 0.0001$, ns $p > 0.05$.

## Overall quality comparison for the theme of GD between Wikipedia and the Baidu Encyclopedia

GD includes dyspepsia (or indigestion), flatulence of the gastrointestinal tract, and functional gastrointestinal disease. Wikipedia scored higher in the total DISCERN scores. Both sites had no obvious discrepancies in the JAMA and GQS scores (Fig. 3A). Further analysis of DISCERN scores found that Section 1, Section 3 and General had a significant difference (Fig. 3B).

## Overall quality comparison for the theme of BD between Wikipedia and the Baidu Encyclopedia

Another subtype of FGIDS is BD, comprising functional gastrointestinal disorders, constipation, functional constipation, neurogenic bowel dysfunction, intestinal malrotation, dysbiosis, small intestinal bacterial overgrowth, splenic flexure syndrome, functional diarrhea, allergic colitis, and intestinal spasm. Wikipedia scored higher in the total DISCERN and JAMA scores (Fig. 4A). For GQS scores, the two sites had no obvious differences. Wikipedia scored remarkably higher in DISCERN Section 1 and 3 scores.

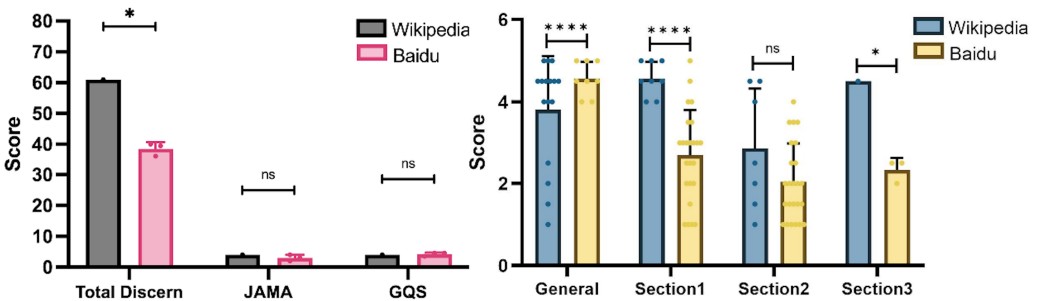

**Figure 3** **Scores in Baidu Encyclopedia and Wikipedia articles of gastroduodenal disorders.** (A) Total DISCERN, JAMA, and GQS scores from the Baidu Encyclopedia and Wikipedia articles. (B) The general and 3 DISCERN section scores. * $p < 0.05$, ** $p < 0.001$, *** $p < 0.0001$, ns $p > 0.05$.

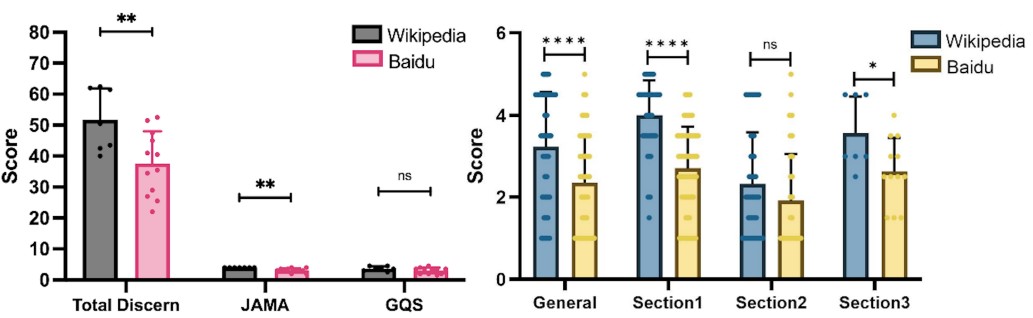

**Figure 4** **Scores in Baidu Encyclopedia and Wikipedia articles on bowel disorders.** (A) Total DISCERN, JAMA, and GQS scores from the Baidu Encyclopedia and Wikipedia articles. (B) The general and 3 DISCERN section scores. * $p < 0.05$, ** $p < 0.001$, *** $p < 0.0001$, ns $p > 0.05$.

## Overall quality comparison for the theme of AD between Wikipedia and the Baidu Encyclopedia

AD includes anal spasm, intestinal dilatation, proctalgia fugax, and levator ani syndrome. Wikipedia scored higher in the total DISCERN and JAMA scores on AD articles. For GQS scores, the two sites had no obvious differences (Fig. 5A). Moreover, there were obvious section differences between the two sites in the DISCERN scores and the General scores (average scores of three sections). This discrepancy was mainly derived from Section 1, indicating more reliability in Wikipedia articles.

## Distribution of the DISCERN scores

Wikipedia's overall quality for total diseases, IBS, GD, BD, and AD was generally higher than Baidu Encyclopedia's, but some scores only differed numerically and were not statistically significant. Hence, we further investigated the score distributions for the two sites. As mentioned above, we considered a score above 3 (60%) points as an indicator of good quality. Table 1 shows how the scores for each section were distributed. Wikipedia had more General DISCERN and Section 1 scores above 3 points, either in overall disease or in IBS, GD, BD, AD, respectively. The Baidu Encyclopedia was lacking in information for

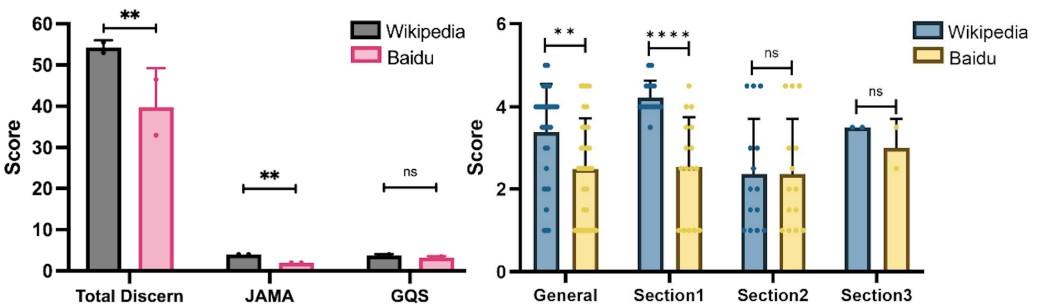

**Figure 5  Scores for Baidu Encyclopedia and Wikipedia articles on anorectal disorders.** (A) Total DIS-CERN, JAMA, and GQS scores from the Baidu Encyclopedia and Wikipedia articles. (B) General scores and 3 DISCERN section scores. * $p < 0.05$, ** $p < 0.001$, *** $p < 0.0001$, ns $p > 0.05$.

**Table 1  Distribution of the DISCERN scores for each disease and comparisons *via* the Fisher's exact test from Wikipedia and the Baidu Encyclopedia.**

| Category | Discern section | General | | | Section 1 | | | Section 2 | | | Section 3 | | |
|---|---|---|---|---|---|---|---|---|---|---|---|---|---|
| | | <3 | ≥3 | p | <3 | ≥3 | p | <3 | ≥3 | p | <3 | ≥3 | p |
| All diseases | Wikipedia | 53 | 123 | <0.0001 | 4 | 84 | <0.0001 | 48 | 29 | 0.1803 | 1 | 10 | 0.0031 |
| | Baidu | 227 | 141 | | 97 | 87 | | 115 | 46 | | 15 | 8 | |
| IBS | Wikipedia | 2 | 14 | <0.0001 | 0 | 8 | 0.0008 | 2 | 5 | 0.0841 | 0 | 1 | 0.4286 |
| | Baidu | 64 | 32 | | 31 | 17 | | 29 | 13 | | 4 | 2 | |
| GD | Wikipedia | 4 | 12 | 0.0206 | 0 | 8 | 0.0292 | 4 | 3 | 0.6465 | 0 | 1 | 0.2500 |
| | Baidu | 29 | 19 | | 11 | 13 | | 15 | 6 | | 3 | 0 | |
| BD | Wikipedia | 39 | 73 | <0.0001 | 4 | 52 | <0.0001 | 34 | 15 | 0.6888 | 1 | 6 | 0.1473 |
| | Baidu | 115 | 77 | | 46 | 50 | | 62 | 22 | | 7 | 5 | |
| AD | Wikipedia | 8 | 24 | 0.0107 | 0 | 16 | 0.0008 | 8 | 6 | >0.9999 | 0 | 2 | >0.9999 |
| | Baidu | 19 | 13 | | 9 | 7 | | 9 | 5 | | 1 | 1 | |

**Notes.**
All diseases including IBS, GD, BG, and AD.
IBS, Irritable Bow syndromes; GD, Gastroduodenal disorders; BD, Bowel disorders; AD, Anorectal disorders.

some GD and BD characteristics in terms of epidemiology, pathophysiology, prevention, reference, and even some core subjects such as examination, diagnosis, and treatment. For example, the term "functional diarrhea" was lacking. The Baidu Encyclopedia also had a much longer update interval than Wikipedia. However, both sites had less Section 2 scores above 3 points for IBS, GD, BD, and AD. Wikipedia had more Section 3 scores above 3 points in the overall disease. Likewise, the two sites had no difference in the proportion of Section 3 scores above 3 points in IBS, GD, BD, and AD respectively, like "功能性腹泻" (functional diarrhea) in the Baidu Encyclopedia, which rated "good quality" (Table 1).

## DISCUSSION

### Principal findings

Over the last two decades, patients have been increasingly using the Internet to seek diagnosis and treatment information for diseases, especially for functional disorders

such as IBS and constipation, which lack biomarkers that are universally accepted for diagnosing these conditions and whose diagnosis depends on self-reported symptom clusters. Most IBS patients do not receive adequate education from doctors due to its complex pathophysiology and limited time in the outpatient setting. This leads them to seek information online. In the general population, self-reported IBS has triple the prevalence compared to clinical diagnostic criteria (*Van den Houte et al., 2019*). An online search helps patients decide when to see a doctor and may improve their health outcomes and care quality. However, false or misleading information may cause misdiagnosis, treatment delays, and serious health risks (*Suarez-Lledo & Alvarez-Galvez, 2021*). Online resources vary in their credibility, accuracy, and readability. Evaluating these aspects is urgent, especially for multiple gastroenteric malignancy mimic disorders. Our research compared the Baidu Encyclopedia and Wikipedia on the reliability, readability, and objectivity of their articles about total diseases, IBS, GD, BD, and AD. Compared to the Baidu Encyclopedia, Wikipedia clearly scored higher in total DISCERN, DISCERN Section 1 and Section 3; there were more total DISCERN and Section 1 scores based on the "above 3 (60%) rule"; and it scored higher in JAMA, suggesting it might provide more reliable, higher quality, and more objective information; and it scored higher in GQS, suggesting it might be better flow or easier to apply. Most DISCERN Section 2 scores distribution for both sites were below 3 points, suggesting a similar and mediocre influence on patients' treatment choices.

According to surveys, 239.33 million people in China (*Yacob et al., 2020*), and half of American adults (*Zhao & Zhang, 2017*) used online resources for medical information. However, online information quality about IBS varies widely (*Levine et al., 2020*). A total of 3%–89% of the retrieved IBS content was found to be relevant to IBS (*Cruz et al., 2019*), with similar problems in the health contents in Wikipedia (*Levine et al., 2020*) and the Baidu Encyclopedia (*Ma et al., 2022*). Wikipedia excels in accuracy, breadth, depth, and informativeness for GI diseases (containing IBS, GD, BD, and AD), which is displayed in Figs. 1–5 (*Good et al., 2012*). One possible reason for the accuracy gap between the two sites is that Baidu Encyclopedia does not have a well-developed consultation system to address issues that may arise from editorial mistakes (*Zhenyu He, 2019*; *Jiang et al., 2022*). Editors may be motivated by personal interests and information may not be rigorously vetted. Wikipedia offers a talk page for editors to discuss and correct errors, which enhances its accuracy (*Shang, 2018*). Secondly, the breadth gap found in the Baidu Encyclopedia may be due to its poor organization and single classification, while Wikipedia has multiple classifications (*Shang, 2018*). Thirdly, there is a talk page in Wikipedia, which enables editors to exchange information and facilitate the participation of editors with higher academic achievement (*Heilman & West, 2015*). The academic weakness of modern medicine in China also leads to less valuable Chinese sources for editors. Fourthly, the informativeness gap is partly due to the motivation of the editors. Editors who work for traditional media tend to omit information that is not relevant to their interests, while Wikipedia editors can use the talk page to facilitate the addition of information. Traditional Chinese medicine lacks sufficient studies in the Western world because of the scarcity of ancient books and the language barrier. This hinders Wikipedia's performance on this topic. The interpretation

of these four features are consistent with the DISCERN scores, and indicate that Wikipedia publications have more reliability and a better impact on patients' treatment choices. It was gratifying that recent updates to the Baidu Encyclopedia have increased the involvement of medical professionals in reviewing or writing the content, which enhances the DISCERN and JAMA scores significantly.

Accurate disease information found in online encyclopedias can enable patients to gain a comprehensive understanding of medical information and avoid misdiagnosis and treatment delays. The diagnosis of IBS is primarily based on symptoms, and the wide range of symptoms increases the probability of a misdiagnosis (*Lacy et al., 2021*). Hence, patients without a clear diagnosis of IBS are more likely to look up their symptoms online (*Van den Houte et al., 2019*). Our research showed that a differential diagnosis or possible overlapping diagnosis of IBS was not mentioned in the Baidu Encyclopedia, however, it was listed briefly in Wikipedia (*Ford et al., 2020*). For example, one disease of GD, "消化不良" (indigestion or dyspepsia), lacks a differential diagnosis and management in primary care in both sites. However, the guidelines emphasized differentiating between dyspepsia and IBS with regard to the hierarchical diagnosis and treatment of dyspepsia (*Black et al., 2022*). Therefore, both sites had poor scores in JAMA, GQS and Section 2 DISCERN for GD. For BD, with regard to "小肠菌群过度生长综合征" (small intestinal bacterial overgrowth) and "神经源性肠道" (neurogenic bowel), Wikipedia showed detailed content on the definition, diagnosis and treatment, while the Baidu Encyclopedia only presented the definition of the two entries. For AD, both sites had little information on treatment and differential diagnosis, such as "肛提肌综合征" (levator ani syndrome). The lack of information may result from the need to search for a very specific disease name "肛提肌综合征" (levator ani syndrome) for which there is little awareness.

By comparing IBS, GD, BD, and AD content on both sites by the above 3 (60%) rule, Wikipedia content generally received "good quality" ratings, whereas the Baidu Encyclopedia content mostly got "fair quality" ratings. For example, both reviewers rated "功能性腹泻" (functional diarrhea) in the Baidu Encyclopedia as "poor quality". The web page for "functional diarrhea" lacked essential information on epidemiology, pathophysiology, prevention, and reference. It also omitted most of the core content for examination, diagnosis, and treatment and did not provide any precautions for drug use, such as drug contraindications. Furthermore, "functional diarrhea" had a much longer update interval in the Baidu Encyclopedia than in Wikipedia. The former updated this entry on October 3, 2017, while the latter did so on December 13, 2022. In addition, the former included no pictures, while the latter had four, indicating Wikipedia was more informative in general. Hence, measures should be taken to enhance content regulation, provide for a review mechanism for interest-related content, and to update the content of IBS articles promptly in the Baidu Encyclopedia.

In the Baidu Encyclopedia, reviewers rated the information for "肠易激综合征" (irritable bowel syndrome) and "功能性腹泻" (functional diarrhea) as "good quality". The good quality rating is possibly due to the high quality of the content provider. Most providers came from unregistered individuals in the past, whose contributions often lack references, in-depth information, and quality in the Baidu Encyclopedia. We also found

only one advertising link on the two web pages, and the article content was objective and scientific, this is superior to the previous report: more than 25 irrelevant advertising links on a Baidu Encyclopedia web page, and the article was biased and unscientific (*Ma et al., 2022*). This improvement might be a result of the Baidu Encyclopedia "rainbow plan", which restricts the editing and revision of medical entries to certified medical experts (*Beijing News, 2023*). Hence, more medical professionals may be motivated or invited to edit or proofread the disease content in the Baidu Encyclopedia, ensuring objectivity and comprehensiveness.

Most Wikipedia articles were close to "good quality" but not "excellent quality", and required more improvement. Louis Levine revealed that Wikipedia was the most commonly viewed of 14 websites in searches for IBS, the highest scoring resource for relevant IBS content, it scored higher (74.3 out of 93) than other websites (*Suarez-Lledo & Alvarez-Galvez, 2021*). It has two content quality evaluation systems: the "Wiki-Project article quality grading scheme" and "Wiki-Project priority assessments". However, these systems cannot fully address the challenges of content accuracy and readability that Wikipedia faces (*Wikipedia, 2023b*; *Wikipedia, 2023c*). To enhance accuracy, it may benefit from more expert input, diverse edits and external collaborations (*Shafee et al., 2017*; *Ma & Zhang, 2020*; *Wang et al., 2020*). To improve the experience of new editors and external collaborations, encyclopedia norms and bureaucracy should be clearer and more straightforward.

Based on the theory that the higher the content score of the resource, the more difficult it is to read (*Wong & Cheung, 2019*), Wikipedia health content needs to be simple and easy to understand. For instance, one effective diet treatment for IBS is described as "a low FODMAPs diet" in Wikipedia, it explains "a low FODMAPs diet" as a type of short-chain carbohydrate that the small intestine absorbs poorly. The explanation seems obscure and professional. While the Baidu Encyclopedia explains that "减少产气食物 (奶制品，大豆，扁豆等)" (reduces gas-producing foods (dairy, soybeans, lentils)), which seems easy to understand. Hence, automatic feedback on the readability can make Wikipedia more user-friendly to patients.

## LIMITATIONS

Our research has some limitations. We only analyzed the information on IBS, GD, BD and AD from the Baidu Encyclopedia and Wikipedia as of October 21, 2022. However, online information changes rapidly, so our results may not reflect the current situation. To ensure the validity and accuracy of our findings, we need to update the data regularly. To evaluate website information quality, tools such as DISCERN, JAMA, and GQS were used. However, tools to estimate the difficulty of reading passages are also needed, such as the Flesch Reading Ease Score scale, which evaluates readability by considering word length, syllable count, and number of sentences. Patients and physicians may have different expectations regarding website content. To design balanced websites that are easily readable by the public and contain the most important content, doing so may increase patients' understanding of IBS and improve health outcomes for IBS patients.

## CONCLUSIONS

Online encyclopedias help users to access disease information and decide when to seek treatment. Wikipedia has more formal development, provides more reliable, higher quality, and more objective IBS-related health information than the Baidu Encyclopedia. Both sites have comparable usability and user experience. Both sites still have much room to improve their information quality. To foster a robust and sustainable online medical system, medical professionals and institutions should collaborate with online platforms to offer reliable, accessible, and clear disease information to the public.

**Abbreviations**

| | |
|---|---|
| **IBS** | irritable bowel syndrome |
| **FGID** | functional gastrointestinal disorder |
| **GD** | gastroduodenal disorders |
| **BD** | bowel disorders |
| **AD** | anorectal disorders |
| **JAMA** | Journal of the American Medical Association |
| **GQS** | Global Quality Score |

## ACKNOWLEDGEMENTS

We thank Dr. Shan-Zun Wei (The Second Affiliated Hospital of Nanchang University, Nanchang, CHN) for giving some advice about data analysis.

### Funding

This work was supported by the Science and Technology Foundation of Sichuan Province of China (No. 2023YFS0287). The funders had no role in study design, data collection and analysis, decision to publish, or preparation of the manuscript.

### Grant Disclosures

The following grant information was disclosed by the authors:
Science and Technology Foundation of Sichuan Province of China: 2023YFS0287.

### Competing Interests

The authors declare there are no competing interests.

### Author Contributions

- Xi Li conceived and designed the experiments, performed the experiments, analyzed the data, prepared figures and/or tables, authored or reviewed drafts of the article, and approved the final draft.
- Kexin Chen conceived and designed the experiments, analyzed the data, prepared figures and/or tables, and approved the final draft.

- Yongbin Jia conceived and designed the experiments, performed the experiments, analyzed the data, prepared figures and/or tables, authored or reviewed drafts of the article, and approved the final draft.
- Fang Yin performed the experiments, prepared figures and/or tables, and approved the final draft.
- Xi Wen conceived and designed the experiments, analyzed the data, authored or reviewed drafts of the article, and approved the final draft.
- Chunhui Wang conceived and designed the experiments, authored or reviewed drafts of the article, and approved the final draft.
- Zhipeng Li conceived and designed the experiments, performed the experiments, analyzed the data, prepared figures and/or tables, authored or reviewed drafts of the article, and approved the final draft.
- Hu Zhang conceived and designed the experiments, prepared figures and/or tables, authored or reviewed drafts of the article, and approved the final draft.

## Data Availability

The raw data is available in the Supplemental Files.

## Supplemental Information

Supplemental information for this article can be found online at http://dx.doi.org/10.7717/peerj.17264#supplemental-information.

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
