# Peer review of "Assessment of medical information on irritable bowel syndrome information in Wikipedia and Baidu Encyclopedia: comparative study"

_PeerJ, doi:10.7717/peerj.17264_

## Round 0.1 · original submission · Minor Revisions

The author is requested to revise the manuscript according to the opinions of the two reviewers.

**Language Note:** The review process has identified that the English language must be improved. PeerJ can provide language editing services - please contact us at [email protected] for pricing (be sure to provide your manuscript number and title). Alternatively, you should make your own arrangements to improve the language quality and provide details in your response letter. – PeerJ Staff

·

Basic reporting

1) Although the general quality of the language is adequate, tI think that it could still be improved in places to ensure that an international audience can clearly understand the text. I’ve listed multiple examples where the language could be improved (including lines 7, 28 , 205, 218, 267) because the current phrasing makes comprehension difficult at times. I suggest you have a colleague who is proficient in English and familiar with the subject matter review your manuscript, or contact a professional editing service. There are also numerous typos and copy editing issues throughout the text which should be improved before acceptance. For example, in line 170, “figure” is misspelled as “figue”. In line 222, the first word of the sentence is not capitalized.

2) The figures could be improved to better convey the data. Because the main statistical comparisons are made between Wiki and Baidu, the bar graphs of these values should be next to each other on the charts. To achieve that, the tests (JAMA, GQS, etc) should be placed on the x-axis and Wiki and Baidu should be placed in the chart legend. This will also make denoting the statistical differences easier as the lines would not need to cross through other bars in the chart.

3) There was no discernable hypothesis stated before the results of the study were given. It is unclear whether the goal of the study was to determine whether there was a difference between the encyclopedias or whether the articles exhibited quality.

Experimental design

1) In general, I find that the methods are defined in sufficient detail to replicate the studies and the methods were applied rigorously to yield legitimate results. Additionally, the topic is related to health sciences, fitting the scope of the journal. Finally, evaluating the reliability, readability, and objectivity of online IBS information is a research objective that is clear and unambiguous. I have no issues with the Experimental design section.

Validity of the findings

1) The accuracy of online information is an important topic to the increasingly online worldwide populations. I commend the authors for their novel study and I believe that the findings can be used to set a baseline for whether future policy changes will improve these results.

2) Although the primary data was provided, navigating the files was challenging due to the frequent use of Chinese characters. Is there a way to add columns for translations? Otherwise, I do not know how these data can be verified for those who are not bilingual.

Reviewer 2 ·

Basic reporting

This is a helpful review and comparison of two online encyclopedias; the Wikipedia used for the English-spoken population and the Baidu for the Chinese-spoken population. This forms an important study as it highlights its role for professionals seeking and/or providing information and for the lay population that may use this as part of patients' empowerment and education.

However, although the title of the study and the abstract refer to information on IBS, in the introduction session the 'aims' change into assessment of the content of the above mentioned encyclopedias for IBS and other FGIDs to include GD, BD and AD. The data and results are presented for all these entities. The authors may need to explain why did they opt to include all of the above FGIDs or correct the title and abstract to reflect this.

A typo error: line 170 Figure 1-5
line 206-207: the information absence...
It is not clear what the authors mean in this sentence: Is it the absence of information? What do they mean by 'strong professionalism' and 'little awareness'? Perhaps rephrasing this for better clarification.

Experimental design

This article is original in comparing the two encyclopedias regarding FGIDs; comparisons have been made in the past about other different topics but not in FGIDs.

As mentioned above it may need to be clarified that the comparison made includes other FGIDs not only IBS.

Methods and data collection were at a very good standard. Figure showed the superiority of the Wikipedia compared to Baidu in all tools and Discern sections; if this is why the authors decided to include all FGIDs this needs to be explained.

I was unable to access the Appendix 1 and info provided; not clear to me whether this was authors' responsibility rather than the Journal's.

Validity of the findings

This article presented data information and food for thought for both encyclopedias and scope for improvement.

However, the authors offered more information about certain subsections, i.e. functional diarrhea in the discussion section whereas none of these differences observed were presented in the results section where we would expect to see more detailed presentation and comparison. Similarly, subsections such as SIBO and functional dyspepsia are mentioned in the 'Discussion' section but results are not elaborated in the 'Results' section.

Additional comments

Overall is a good study comparing the two encyclopedias. I would suggest more detailed account of the relevant subgroups in the result section so discussion becomes more relevant to the reader.

---

## Round 0.2 · Minor Revisions

The authors need to make revisions as requested by the reviewer.

·

Basic reporting

1) Although the general quality of the language is adequate, a few changes are needed.
Line 64. “US” should be written as the “United States”.
Line 85. “functional constipation, and so on” should be “and functional constipation” to remove the informal language.
Line 203. “Table1” should have a space and be written as “Table 1”
Line 315. Parentheses used inside another set of parentheses should be represented as brackets and the informal “etc” should be removed. The text should read “(reduces gas-producing foods [dairy, soybeans, and lentils])”
Figure legends. Not all legends state what the asterisks represent in the corresponding figure. Please add this to every figure.
Table 1 footnote: This footnote contains different fonts or font sizes. The footnote of the table should be in a uniform font.

2) The revised figures are a significant improvement on the original figures. However, the colors used in the bar graphs for the B panels should be changed. Currently, they are both similar shades of purple. Colors should be used that have more contrast.

3) I would add the exact search terms used (described in Lines 112-3).

Experimental design

As noted in my original review, I find that the methods are defined in sufficient detail to replicate the studies and the methods were applied rigorously to yield legitimate results.

Validity of the findings

The additional of English translations has improved the excel files. As noted in my original review, I think that the accuracy of online information is an important topic to the increasingly online worldwide populations. I commend the authors for their novel study and I believe that the findings can be used to set a baseline for whether future policy changes will improve these results.

Additional comments

The text has been significantly improved following revisions. However, a few minor revisions are still required.

---

## Round 0.3 · accepted · Accept

Although only one reviewer agreed to receive the manuscript, the revisions suggested by the other reviewer have been refined by the authors. I reviewed the manuscript and determined that there was no obvious risk of publication and that it was worthy of publication; therefore, I approved the manuscript for publication.